# The Impact of Glomerular Disease on Dyslipidemia in Pediatric Patients Treated with Dialysis

**DOI:** 10.3390/nu17030459

**Published:** 2025-01-27

**Authors:** Edward Zitnik, Elani Streja, Marciana Laster

**Affiliations:** 1Department of Pediatrics, University of Connecticut School of Medicine, Farmington, CT 06032, USA; 2Fielding School of Public Health, University of California Los Angeles, Los Angeles, CA 90095, USA; estreja@hs.uci.edu; 3Department of Pediatrics, Indiana University School of Medicine, Indianapolis, IN 46202, USA; mlaster@iu.edu

**Keywords:** End Stage Kidney Disease (ESKD), dialysis, pediatric, dyslipidemia, nutrition

## Abstract

Background/Objectives: Children on dialysis have a 10-fold increase in cardiovascular disease (CVD)-related mortality when compared to the general population. The development of CVD in dialysis patients is attributed to Chronic Kidney Disease–Mineral Bone Disorder (CKD-MBD) and dyslipidemia. While the prevalence of dyslipidemia in adult dialysis patients has been described, there are limited data on prevalence, severity, and risk factors for pediatric dyslipidemia. Methods: Data from 1730 pediatric patients ≤ 21 years receiving maintenance hemodialysis or peritoneal dialysis with at least one lipid panel measurement were obtained from USRDS between 2001 and 2016. Disease etiology was classified as being glomerular (*n* = 1029) or non-glomerular (*n* = 701). Comparisons were made across etiologies using both linear and logistic regression models to determine the relationship between disease etiology and lipid levels. Results: The cohort had a mean age of 15.2 years and were 54.5% female. Adjusting for age, sex, race/ethnicity, modality, time with End Stage Kidney Disease (ESKD), and body mass index (BMI) and using non-glomerular etiology as the reference, glomerular disease [mean (95% CI)] was associated with +19% (+14.7%, +23.8%) higher total cholesterol level (183 mg/dL vs. 162 mg/dL), +21% (+14.8%, +26.6%) higher low density lipoprotein cholesterol level (108 mg/dL vs. 87 mg/dL), and +22.3% (+15.5%, +29.5%) higher triglyceride level (169 mg/dL vs. 147 mg/dL). Glomerular disease [OR (95% CI)] was associated with 3.0-fold [2.4, 3.9] higher odds of having an abnormal total cholesterol level, 3.8-fold [2.8, 5.0] higher odds of having an abnormal LDL-C level, and 1.9-fold [1.5, 2.4] higher odds of having an abnormal triglyceride level when compared to non-glomerular disease. Conclusions: Pediatric dialysis patients have a high prevalence of dyslipidemia, particularly from elevated triglyceride levels. Specifically, patients with glomerular disease have an even higher risk of dyslipidemia from elevated non-HDL cholesterol and triglyceride levels than patients with non-glomerular disease. The long-term impact of this unfavorable lipid profile requires further investigation.

## 1. Introduction

Over half of all deaths among dialysis patients are attributed to cardiovascular causes [1]. The development of cardiovascular disease (CVD) in dialysis patients is attributed to progressive blood vessel calcification from impaired mineral homeostasis, in addition to dyslipidemia [2].

A study of pediatric CKD patients in the United States estimated the prevalence of dyslipidemia in patients with both glomerular and non-glomerular disease to be 43% and 35%, respectively. There was a stronger association with adverse changes in lipid levels over time in children with glomerular disease [3]. The prevalence of dyslipidemia was 61.5% in a study of Korean children less than 20 years old across various etiologies of CKD [4], and very high TC was associated with CKD progression [5]. A small study of 70 pediatric dialysis patients in Mexico found that the baseline prevalence of dyslipidemia was 57% at the start of dialysis initiation [6]. Furthermore, 85% of peritoneal dialysis patients and 76% of hemodialysis patients had dyslipidemia in a cohort of European children [7]. Beyond these studies, most of the research of dyslipidemia and its impact on health outcomes for dialysis patients has been in adult patients.

The available data fail to probe the impact of disease etiology on dyslipidemia risk. Because of factors inherent to glomerular disease, children with glomerular disease demonstrate high rates of dyslipidemia, particularly hypercholesterolemia [8]. At present, little is known about the persistence of glomerular-related dyslipidemia once patients have started dialysis, a time characterized by disease burnout and decreased immunosuppression. While these groups have been compared in the non-dialysis CKD population, less is known about how dyslipidemia compares between disease etiologies in pediatric patients treated with dialysis. Given this, the objectives of this study are to define the prevalence of dyslipidemia and to compare the risk of dyslipidemia for glomerular and non-glomerular etiologies of ESKD.

## 2. Materials and Methods

### 2.1. Population

This is a retrospective cohort study of 1730 pediatric patients aged 0–21 years old from the United States Renal Disease Systems (USRDS) who were treated with maintenance hemodialysis or peritoneal dialysis between 2001 and 2016. Included participants had at least one lipid panel measurement within a year of dialysis initiation and a known etiology of their kidney failure, categorized as either glomerular or non-glomerular. Excluded patients included those who lacked lipid data, had a first lipid panel drawn more than one year from the dialysis start date, were older than 21 years old, entered USRDS as a transplant patient, were missing a dialysis start date, or did not have a known etiology for their End Stage Kidney Disease (ESKD). Repeated lipid panel measurements were excluded, and the observation closest to dialysis initiation was analyzed. A cohort construction is shown in Appendix A.

### 2.2. Patient Data

Patients were classified into two major disease etiology categories: glomerular etiology (*n* = 1029) and non-glomerular etiology (*n* = 701). Consistent with the classifications set forth by prior pediatric studies of glomerular disease, glomerular disease included focal segmental glomerulosclerosis (FSGS), nephrotic syndrome, hemolytic uremic syndrome (HUS), and other forms of glomerulonephritis. Non-glomerular disease included polycystic kidney disease, kidney tumors, congenital anomalies of the kidney and urinary tract (CAKUT), and urologic conditions [9]. Age, sex, race/ethnicity, dialysis modality, insurance status, height, weight, BMI, and serum albumin were available for all patients. Race and ethnicity are reported as Black (non-Hispanic Black), White (non-Hispanic White), Hispanic, Asian, and Other, which includes Native American and unknown race. Racial categorization in the USRDS is based on self-report.

### 2.3. Measures of Dyslipidemia

Total cholesterol, LDL-C, HDL-C, and TG levels are reported as median and interquartile ranges (IQRs) given their nonsymmetric distribution. Extreme values (TC ≥ 700 mg/dL, LDL-C ≥ 600 mg/dL, TG ≥ 2000 mg/dL, and HDL-C ≥ 100 mg/dL) were excluded, as these data may represent implausible values or patients with a primary lipid disorder, such as familial hypercholesterolemia. These cutoff points were extrapolated from the lipid levels, which may be consistent with a primary lipid disorder per the National Heart, Lung, and Blood Institute (NHLBI) [10]. The lipid values were also categorized into acceptable, borderline, and abnormal groups based on the NHLBI thresholds for patient ages, which are shown in Appendix A.

### 2.4. Statistical Analysis

Between-group comparisons were made using the Wilcoxon Rank Sum test for medians (IQR) and the chi-square test for comparisons of proportions. Simple linear regression models were created to determine the relationship between disease etiology and each lipid measure (TC, LDL-C, HDL-C, and TG). Lipid measures were log-transformed for use in linear regression models given their nonsymmetric distribution and are presented as the percent difference between each group using non-glomerular disease as the reference. Percent differences were calculated according to the formula [(10^B^) − 1] × 100. Models were adjusted for age, sex, race, dialysis modality, insurance status, and time with ESKD. Logistic regression models were used to determine the relationship between disease etiology and the odds of abnormal lipids (elevated TC, elevated LDL-C, elevated TG, and decreased HDL-C) in each individual dyslipidemia category defined by age-specific NHLBI thresholds. As above, models were adjusted for age, sex, race, dialysis modality, and time with ESKD. A *p* value < 0.05 denoted significance for all analyses. All analyses were completed in SAS Studio version 3.81.

## 3. Results

### 3.1. Cohort Comparisons

Characteristics of the cohort are presented in Table 1. Patients with a glomerular etiology for their ESKD were older (17.7 vs. 15.2 years), taller (162 cm vs. 151 cm), and had a higher BMI (22.4 vs. 19.9) when compared with patients with a non-glomerular etiology for their ESKD. There was a relatively even distribution of male and female subjects in the glomerular group (50.8% and 49.2%) and more males than females in the non-glomerular group (59.8% and 40.2%). Almost half of all patients with non-glomerular disease were White (45.4% vs. 28.6%), whereas the glomerular group had a greater prevalence of Black participants (32.6% vs. 20.0%) than the non-glomerular group. There was no difference in dialysis modality between the groups, and the majority of all patients were treated with hemodialysis regardless of ESKD etiology (73.2% for the glomerular group and 73.0% for the non-glomerular group). Median lipid levels by sex and age are shown in Appendix A.

### 3.2. Linear Analysis of Lipid Parameters

Median (IQR) lipid levels were compared between the groups. Unadjusted median levels of total cholesterol, LDL-C, and triglyceride were all higher in the glomerular group than in the non-glomerular group, as shown in Table 2. The median (IQR) total cholesterol level for the glomerular group was 183 mg/dL (146, 232) vs. 162 mg/dL (130, 192) for the non-glomerular group. The median (IQR) LDL-C level for the glomerular group was 108 mg/dL (78, 140) vs. 87 mg/dL (65, 111) for the non-glomerular group. The median (IQR) triglyceride level for the glomerular group were 169 mg/dL (116, 242) vs. 147 mg/dL (102, 223) for the non-glomerular group. These differences were statistically significant, with *p* < 0.0001 for all three parameters. There was no statistical difference detected between the two groups for HDL-C level. In linear regression models adjusted for age, sex, race, dialysis modality, insurance status, and time with ESKD, glomerular disease was associated with a percent difference (95% CI) of +19% (+14.7%, +23.8%) higher total cholesterol level, +21% (+14.8%, +26.6%) higher LDL-C level, and +22.3% (+15.5%, +29.5%) higher triglyceride level (*p* < 0.0001 for all) than non-glomerular disease. There was no significant difference between the two groups for HDL-C level.

### 3.3. Categorical Analysis of Lipid Parameters Using NHLBI Definitions

The prevalence of dyslipidemia in the entire cohort and by etiology of ESKD is shown in Table 3. Dyslipidemia, defined as one or more abnormal lipid parameters, was present in 85.5% of all pediatric dialysis patients, 87.9% of patients with glomerular disease, and 82.0% of patients with non-glomerular disease. More than 60% of all patients had dyslipidemia from abnormal triglyceride levels and more than 50% of all patients had dyslipidemia from abnormal HDL-C levels.

Table 4 displays the proportion of patients with dyslipidemia for each lipid parameter by etiology of ESKD. The prevalence of dyslipidemia for patients with ESKD from glomerular causes and non-glomerular causes was 37.8% vs. 20.0% for elevated total cholesterol level (*p* < 0.001), 33.1% vs. 12.8% for elevated LDL-C level (*p* < 0.001), 64.7% vs. 60.0% for elevated triglyceride level (*p* = 0.049), and 54.5% vs. 54.4% for decreased HDL-C level (*p* = 1). There was no statistical difference detected between the two groups for triglyceride and HDL-C levels in unadjusted analysis. In the logistic regression analysis, glomerular disease [OR (95% CI)] was associated with 3.0-fold [2.4, 3.9] higher odds of having an elevated total cholesterol level (*p* < 0.0001), 3.8-fold [2.8, 5.0] higher odds of having an elevated LDL-C level (*p* < 0.0001), and 1.9-fold [1.5, 2.4] higher odds of having an elevated triglyceride level (*p* < 0.0001). There was no significant difference in the odds of a decreased HDL-C level. These results are further delineated in Figure 1, which stratifies each group into acceptable, borderline, and abnormal lipid levels.

To assess the patterns of significant proteinuria in dyslipidemia, the cohort was stratified by normal serum albumin (≥3.5 g/dL) and low serum albumin (<3.5 g/dL). Figure 2 displays a cross-section of low and normal serum albumin and absence and presence of dyslipidemia, stratified by etiology of ESKD. Of all patients with glomerular disease and dyslipidemia, 68% had low serum albumin compared to 40% of the non-glomerular group with dyslipidemia. Therefore, amongst those with dyslipidemia, concurrent hypoalbuminemia was more frequent in the glomerular group. In those without dyslipidemia, 53% of the glomerular group had low serum albumin compared to 43% of the non-glomerular group. Thus, even in those without dyslipidemia, hypoalbuminemia was more common in the glomerular group.

## 4. Discussion

Dyslipidemia, defined as having an abnormal value for at least one lipid parameter (elevated TC, elevated LDL-C, elevated TG, or decreased HDL-C), was prevalent across the entire cohort of pediatric patients treated with dialysis, with 85.5% of all dialysis patients affected by dyslipidemia. Although the prevalence was slightly higher in glomerular disease (87.9%), dyslipidemia was highly prevalent in the non-glomerular disease population, as well (82%). The most common abnormal lipid parameter was hypertriglyceridemia, with 61.2% of all dialysis patients affected, equating to 64.7% of patients with glomerular disease and 60.1% of patients with non-glomerular disease. Patients with ESKD from glomerular causes were found to have higher levels of total cholesterol, LDL-C, and triglyceride than patients with ESKD from non-glomerular causes. HDL-C levels were nearly identical between the groups. When categorized by the age-based thresholds set by the NHLBI, patients with ESKD from glomerular causes were found to have higher odds of dyslipidemia based on elevated total cholesterol levels, elevated LDL-C levels, and elevated triglyceride levels than patients with ESKD from non-glomerular causes. Thus, pediatric dialysis patients experience dyslipidemia at a high rate, and the risk of dyslipidemia is even greater in those with glomerular disease as the cause of ESKD.

Classically, dyslipidemia in CKD is characterized by elevated triglyceride and low HDL cholesterol levels owing to decreased renal clearance and decreased activity of enzymes like lipoprotein lipase. Although major increases in LDL-C tend to be absent even in the adult CKD population, there exists a tendency toward a greater prevalence of atherogenic small, dense LDL-C particles and accumulation of atherogenic proteins, such as apolipoprotein B, in CKD [11]. Additionally, other unmeasured pro-atherogenic properties of LDL-C include its oxidation due to chronic inflammation. Given this, the importance of LDL-C in cardiovascular outcomes in general, and in particular in the glomerular disease population, cannot be overshadowed by the low prevalence of abnormal LDL-C based on routinely measured LDL-C levels. Although the overall prevalence of elevated LDL-C was only 24.9% in our cohort, the risk of having an elevated LDL-C was significantly greater in the glomerular group compared to the non-glomerular group. It will be critical that future studies evaluate the impact of atherogenic qualities of LDL-C, as opposed to simple LDL-C levels, on cardiovascular outcomes in children.

Glomerular disease associated with nephrotic range proteinuria carries an additional risk of elevated LDL-C levels owing to impaired clearance of lipoproteins as a result of decreased hepatic and peripheral lipoprotein lipase and increased cholesterol biosynthesis [12]. The importance of urinary protein loss in dyslipidemia is underscored by Saland et al., who demonstrated that an increase in proteinuria over time was associated with an increase in non-HDL cholesterol and triglyceride levels regardless of CKD etiology [3]. Although this association was of greater magnitude in the glomerular etiology cohort, the association held even for non-glomerular diseases. Given that patients in remission (decreased loss of urinary protein) typically have normal lipid levels, the degree to which this mechanism contributes to dyslipidemia upon dialysis initiation, when urine output and thus urinary protein loss are often decreasing, is unknown; however, the differences observed between the two groups suggest the persistence of a physiologic role of the glomerulus in lipid metabolism even after starting dialysis.

One possible explanation for why patients with glomerular disease have worse dyslipidemia than patients with non-glomerular disease is differences in podocyte function. Junctional adhesion molecule-like protein (JAML) is expressed in the podocyte in higher amounts in mice with proteinuria from glomerular disease. JAML expression was also associated with lipid accumulation. This podocyte-specific protein appears to play a role in lipid metabolism at the level of the glomerulus and may explain the observed difference between the two groups in this study [13].

The findings of this study, particularly the increased risk of dyslipidemia in the glomerular population even after dialysis initiation, have potential implications for the care of children on dialysis. Unfortunately, treatment of the lipid disturbances observed in pediatric patients with CKD has remained controversial due to lack of data supporting its safety and efficacy. Kidney Disease Improving Global Outcomes (KDIGO) [14] recommends initiation of a statin or statin/ezetimibe combination for adult non-dialysis CKD patients based on age, comorbidities, and estimated 10-year risk of coronary death or non-fatal myocardial infarction. In pediatric CKD patients, KDIGO recommends yearly lipid panel screening but recommends against initiating lipid-lowering medication in children with CKD of any stage, including those on dialysis, irrespective of risk factors for cardiovascular disease. Still, they do recommend that a child already started on a lipid-lowering agent prior to starting dialysis may continue the medication after dialysis initiation. The KDIGO work group on lipids calls for additional research that can contribute to improved guideline development, highlighting the potential impact of this and future studies in the pediatric dialysis population. While further studies are needed to define the impact of dyslipidemia on well-defined cardiovascular outcomes in pediatric CKD, we do know that there are intermediate outcomes that support an adverse impact of dyslipidemia. A study of non-dialysis CKD patients has shown that increased carotid intima-media thickness, a consequence of dyslipidemia and CKD–Mineral Bone Disorder (CKD-MBD), is already present during childhood in pediatric patients with CKD [15]. Additionally, lipid control may have implications for progression to dialysis in pre-dialysis CKD patients. Baek et al. found that patients with dyslipidemia had faster progression of CKD [5].

Establishing the prevalence and severity of dyslipidemia and its risk factors can provide a basis to study the safety and efficacy of statins, fibrates, fish oil, ezetimibe, or other agents for pediatric dialysis patients with dyslipidemia. A small study of pediatric non-dialysis CKD patients revealed that atorvastatin 10–20 mg daily was safe and effective at lowering LDL-C levels over a 24-week period. There were no episodes of significant transaminitis or myositis reported, with a 40% reduction in LDL-C levels during the study period [16]. This provides hope that pharmaceutical intervention may be possible in this unique patient population. While studies like this are promising, more research is needed to allow for safe and evidence-based interventions for dyslipidemia in this population.

One limitation of this study is the lack of longitudinal lipid data to determine whether the lipid profiles of glomerular and non-glomerular patients remain different over time. Still, this study takes advantage of a large pediatric dialysis population and serves as the basis for future large studies that can assess lipid measurements longitudinally. A longer study period would also help to establish the effects of dyslipidemia on mortality and cardiovascular outcomes. Another limitation of this study is the use of both fasting and non-fasting lipids, as the USRDS database does not specify the collection method for each lipid panel. Although pediatric studies have shown only slight differences between fasting and non-fasting lipids, this has not been validated in pediatric CKD [17]. Therefore, this further underscores the need for prospective studies that adequately assess lipids in a fasting state or allow for the repeat of abnormal non-fasting lipid panels to confirm abnormalities. Although many patients discontinue immunosuppressants once they have reached ESKD, the impact of treatments on lipid levels may linger and may serve as a potential mechanism for the differences in lipids by disease type. Further studies that account for prior treatments are needed to evaluate medication use as an additional mediator. An additional limitation is the inability to look specifically at the impact of proteinuria on dyslipidemia. Although the sub-analysis by albumin level suggests that not all glomerular disease was associated with hypoalbuminemia (a consequence of significant proteinuria) [18], a significant portion did in fact have low albumin levels. Whether this was related to proteinuria or alternative mechanisms, such as malnutrition, can only be assessed by future studies that measure urine output and urinary protein excretion. Finally, this study can only demonstrate associations between glomerular disease and lipid values in patients on dialysis and does not imply causality.

## 5. Conclusions

Pediatric dialysis patients, regardless of ESKD etiology, have a high prevalence of dyslipidemia, particularly from elevated triglycerides. Children with ESKD from glomerular causes have a higher risk of dyslipidemia from elevated non-HDL cholesterol and triglyceride levels. The long-term impact of this unfavorable lipid profile on cardiovascular outcomes requires further investigation, with a particular focus on whether glomerular disease compounds lipid-related cardiovascular risk. Furthermore, patients with ESKD from glomerular causes may require closer lipid monitoring and advanced dietary modification.

## Figures and Tables

**Figure 1 nutrients-17-00459-f001:**
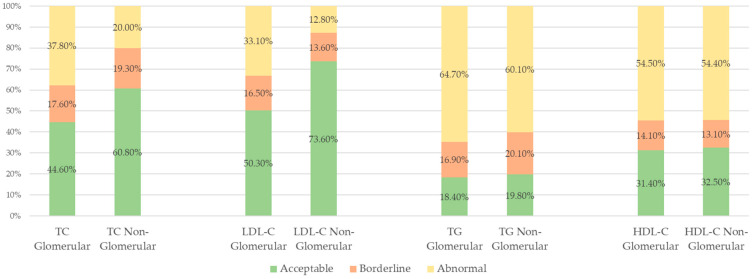
Proportion of dyslipidemia based on ESKD etiology. TC = total cholesterol, LDL-C = low density lipoprotein cholesterol, TG = triglyceride, HDL-C = high density lipoprotein cholesterol.

**Figure 2 nutrients-17-00459-f002:**
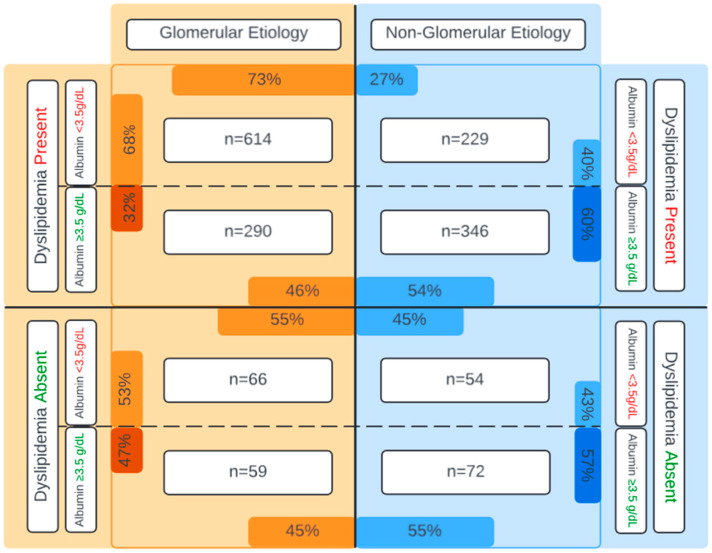
The co-occurrence of hypoalbuminemia with dyslipidemia in glomerular and non-glomerular disease.

**Table 1 nutrients-17-00459-t001:** Cohort characteristics by disease etiology.

	Glomerular Etiology, *n* (%)	Non-Glomerular Etiology, *n* (%)	*p*-Value
*n*	1029	701	
Age, years	17.7 (14.5, 19.7)	15.2 (10.6, 18.7)	<0.0001
Sex, *n* (%)			0.0002
Male	523 (50.8%)	419 (59.8%)	
Female	506 (49.2%)	282 (40.2%)	
Race, *n* (%)			<0.0001
White	294 (28.6%)	318 (45.4%)	
Black	335 (32.6%)	140 (20.0%)	
Hispanic	331 (32.2%)	211 (30.1%)	
Asian	43 (4.2%)	19 (2.7%)	
Other/Unknown/Native American	26 (2.5%)	13 (1.9%)	
Modality, *n* (%)			1
HD	750 (73.2%)	506 (73.0%)	
PD	275 (26.8%)	187 (27.0%)	
Insurance, *n* (%)			0.002
Uninsured	108 (10.5%)	44 (6.3%)	
Public	546 (53.1%)	428 (61.1%)	
Private	260 (25.3%)	160 (22.8%)	
Unknown	115 (11.2%)	69 (9.8%)	
Height, cm	162.0 (151.0, 170.2)	151.0 (126.0, 163.0)	<0.0001
Weight, kg	59.0 (46.0, 77.0)	47.0 (30.0, 63.0)	<0.0001
BMI	22.4 (18.8, 27.8)	19.9 (16.9, 25.1)	<0.0001
Albumin, g/dL	3.1 (2.4, 3.7)	3.7 (3.2, 4.1)	<0.0001

Categorial values are presented as frequency (percentages) and compared using the chi-square test. Continuous variables are presented as the median (interquartile range) and compared using the Wilcoxon Rank Sum test.

**Table 2 nutrients-17-00459-t002:** Unadjusted and adjusted comparison of lipid measures by disease etiology.

Lipid Parameter	Glomerular Etiology, Median mg/dL (IQR)	Non-Glomerular Etiology, Median mg/dL (IQR)	*p*-Value	Linear Regression Models (ref = Non-Glomerular) % Difference (95% CI)
TC	183 (146, 232)	162 (130, 192)	<0.0001	+19% (+14.7%, +23.8%) *p* < 0.0001
LDL-C	108 (78, 140)	87 (65, 111)	<0.0001	+21% (+14.8%, +26.6%) *p* < 0.0001
TG	169 (116, 242)	147 (102, 223)	<0.0001	+22.3% (+15.5%, +29.5%) *p* < 0.0001
HDL-C	38 (30, 49)	38 (30, 50)	0.3	−1.4% (−5.4%, +2.7%) *p* = 0.5

Each lipid parameter is presented as the median value in mg/dL with the interquartile range (IQR) and compared based on the etiology of ESKD. Linear regression models for each lipid parameter are presented as the percentage difference in lipid measurement with 95% confidence intervals using non-glomerular etiology as the reference. Percent difference calculated according to the formula [(10^B^) − 1] × 100. Covariates include race/ethnicity, age, sex, modality, and time with ESKD. TC = total cholesterol, LDL-C = low density lipoprotein cholesterol, TG = triglyceride, HDL-C = high density lipoprotein cholesterol.

**Table 3 nutrients-17-00459-t003:** Proportion of patients with dyslipidemia based on National Heart, Lung, and Blood Institute (NHLBI) cutoff points for age.

	Entire Cohort *n* (%)	Glomerular Etiology *n* (%)	Non-Glomerular Etiology *n* (%)
Any lipid parameter	1479/1730 (85.5%)	904/1029 (87.9%)	575/701 (82.0%)
TC	529/1730 (30.6%)	389/1029 (37.8%)	140/701 (20.0%)
LDL-C	431/1730 (24.9%)	341/1029 (33.1%)	90/701 (12.8%)
TG	1087/1730 (62.8%)	666/1029 (64.7%)	421/701 (60.1%)
HDL-C	942/1730 (54.5%)	561/1029 (54.5%)	381/701 (54.4%)

Data are presented based on proportion of patients meeting diagnostic criteria based on any lipid parameter and each individual parameter. TC = total cholesterol, LDL-C = low density lipoprotein cholesterol, TG = triglyceride, HDL-C = high density lipoprotein cholesterol.

**Table 4 nutrients-17-00459-t004:** Unadjusted and adjusted comparison of the prevalence of lipid measures by disease etiology.

	Glomerular Etiology, *n* (%)	Non-Glomerular Etiology, *n* (%)	*p*-Value	Logistic Regression Models (ref = Non-Glomerular) OR (95% CI)
TC			<0.0001	3.0 (2.4, 3.9) *p* < 0.0001
Acceptable/Borderline	640 (62.2%)	561 (80%)		
Abnormal	389 (37.8%)	140 (20%)		
LDL-C			<0.0001	3.8 (2.8, 5.0) *p* < 0.0001
Acceptable/Borderline	688 (66.9%)	611 (87.2%)		
Abnormal	341 (33.1%)	90 (12.8%)		
TG			0.049	1.9 (1.5, 2.4) *p* < 0.0001
Acceptable/Borderline	363 (35.3%)	280 (40.0%)		
Abnormal	666 (64.7%)	421 (60.0%)		
HDL-C			1	0.9 (0.7, 1.1) *p* = 0.4
Acceptable/Borderline	468 (45.5%)	320 (45.6%)		
Abnormal	561 (54.5%)	381 (54.4%)		

Each lipid parameter is categorized as abnormal or acceptable/borderline based on National Heart, Lung, and Blood Institute (NHLBI) cutoff points for age. Logistic regression models for each lipid parameter are presented as the odds ratio with 95% confidence intervals using non-glomerular etiology as the reference. Covariates include race/ethnicity, age, sex, modality, and time with ESKD. TC = total cholesterol, LDL-C = low density lipoprotein cholesterol, TG = triglyceride, HDL-C = high density lipoprotein cholesterol.

## Data Availability

The raw data supporting the conclusions of this article will be made available by the authors upon request. The dataset was originally obtained from a request to the United States Renal Data System (USRDS).

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
