# Peer review of "The Impact of Glomerular Disease on Dyslipidemia in Pediatric Patients Treated with Dialysis"

_nutrients, 2025, doi:10.3390/nu17030459_

Round 1
Reviewer 1 Report
Comments and Suggestions for Authors
The results of the study from the large-scale data were of value. On the other hand, there appeared to be points to reconsider to increase the quality of the paper.
1. This was a large-scaled study. Many readers would want to see the supplemental data based on the sub-analysis of patients over or under 10 (or 12) year old, for instance. This may be useful for pediatric practice.
2. Many readers would also want to see the supplemental data based on the sub-analysis by sex. This may be useful for sex-based pediatric practice.
3. It is better to add the word ‘cholesterol (or -C) to the terms ‘LDL’ and ‘HDL’ in re-reviewing the overall text.
4. The expression ‘triglyceride’, ‘triglycerides’ or ‘TG abbreviated’ was mixed; then, it could be consistently expressed in re-reviewing the overall text.
5. Abstract; the absolute levels of each lipid concentration in blood should be added in addition to odds ratios.
6. Introduction; how large is the dyslipidemia condition contributory/attributed to cardiovascular morbidity and mortality in children? The concrete data should be summarized or cited.
7. Methods; the measurement methods of lipids (i.e., at lease, assay name) should be described.
8. Methods; the performance of measurement of lipids (e.g., coefficients of variations, reproducibility) should be described.
9. Methods; there was a description of familiar hypercholesterolemia if LDL cholesterol over 600 mg/dL (or total cholesterol over 700 mg/dL). The standard and guideline should be cited.
10. Methods; how was the heterozygotes of familiar hypercholesterolemia treated/operated in the study? The points as suggested in No. 5-8 would be involved in the accuracy of results of the study.
11. Methods; in statistical way, how was the referent group in the logistic regression model considered? The borderline could also include the affected group from the preventative insight.
12. Results; information of used drugs, especially steroids, should be added.
13. The drugs should be added for multivariate analyses.
14. Each Table; the abbreviations could be disclosure in footnote.
15. Discussion; the discussion of mechanistic explanation of results by glomerular etiology could be more detailed. The impact of etiology on the future cardiovascular diseases could be affected and the strategy might be altered by etiology.
16. Discussion; as stated in the study limitation, the sampling methods for lipids were an issue. In fact, how large was the results of the study affected by this limitation? Triglycerides and LDL-cholesterol (if the calculation of LDL-cholesterol included the non-fasting triglyceride levels) could be concretely described.
17. Discussion; as stated in the study limitation, the explanation on the correlation between albumin and proteinuria should be described with any references (please cite evidence).
18. Line 14 (Abstract); patients’ demographics/attributes (i.e., mean age, gender prevalence) could be added.
19. Line 16; the word ‘being’ might be added after the expression ‘classified as’.
20. Line 19; a full spell-out could be required for BMI as abbreviation.
21. The expression ‘higher’ triglyceride in Line 22, ‘abnormal’ triglyceride in Line 24, and T ‘elevated’ triglycerides in Line 26; these appeared to be in a similar context, and the word ‘level’ may be added (e.g., higher level of triglyceride, or higher triglyceride level).
22. Line 38; a wide space was seen before ‘This’.
23. Line 106; a full spell-out could be required for BMI as abbreviation.
24. Line 111; the word of ‘Lipids’ was written in an uppercase/large letter style. There were the same expressions in other parts (for instance, Line 232 etc.). It seemed to be tricky.
25. Line 245; a wide space was seen before ‘Although’.
26. Line 265: ‘,’ might be added after ‘initiation’.
Author Response
Comment 1: This was a large-scaled study. Many readers would want to see the supplemental data based on the sub-analysis of patients over or under 10 (or 12) year old, for instance. This may be useful for pediatric practice.
Comment 2: Many readers would also want to see the supplemental data based on the sub-analysis by sex. This may be useful for sex-based pediatric practice.
Response 1 and 2: This data has been added to the supplement as Supplemental Tables S2 and S3 and referenced in the text as ‘Median lipid levels by sex and age are shown in Supplemental Table S2 and S3.”
Comment 3: It is better to add the word ‘cholesterol (or -C) to the terms ‘LDL’ and ‘HDL’ in re-reviewing the overall text.
Response 3: ‘-C’ added to LDL and HDL, abbreviations also updated in abbreviation list.
Comment 4: The expression ‘triglyceride’, ‘triglycerides’ or ‘TG abbreviated’ was mixed; then, it could be consistently expressed in re-reviewing the overall text.
Response 4: Updated to uniformly use ‘triglyceride’ in the body of the text and ‘TG’ in all tables.
Comment 5: Abstract; the absolute levels of each lipid concentration in blood should be added in addition to odds ratios.
Response 5: Median TC, LDL-C, and TG levels added to abstract and highlighted.
Comment 6: Introduction; how large is the dyslipidemia condition contributory/attributed to cardiovascular morbidity and mortality in children? The concrete data should be summarized or cited.
Response 6: Added to introduction: ‘There are two seminal studies which establish the relationship between dyslipidemia and cardiovascular mortality in children. The first is the Bogalusa Heart Study which began in 1972. It showed that atherosclerotic lesions of the aorta and coronary vessels which were obtained from autopsy samples were present in children with obesity, hypertension, elevated low density lipoprotein cholesterol (LDL-C) and decreased high density lipoprotein cholesterol (HDL-C) [5]. The second is the Cardiovascular Risk in Young Finns Study which began in 1980. It showed that pediatric obesity, hypertension, and dyslipidemia strongly predicted atherosclerosis in adulthood when examined using ultrasound of the carotid and brachial arteries [6].’
Comment 7: Methods; the measurement methods of lipids (i.e., at lease, assay name) should be described.
Comment 8: Methods; the performance of measurement of lipids (e.g., coefficients of variations, reproducibility) should be described.
Response 7 and 8: The USRDS is a database of dialysis data obtained from dialysis centers across the United States. Centers simply report the values and the lab techniques used to measure these values are not known to the database.
Comment 9: Methods; there was a description of familiar hypercholesterolemia if LDL cholesterol over 600 mg/dL (or total cholesterol over 700 mg/dL). The standard and guideline should be cited.
Response 9: Citation added ‘These cut points were extrapolated from the lipid levels which may be consistent with a primary lipid disorder per the National Heart, Lung, & Blood Institute (NHLBI) [10].’
Comment 10: Methods; how was the heterozygotes of familiar hypercholesterolemia treated/operated in the study? The points as suggested in No. 5-8 would be involved in the accuracy of results of the study.
Response 10: It was not known if any patients in the study had a primary lipid disorder such as familial hypercholesterolemia. To reduce the chance of accidentally including a patient with a primary lipid disorder, we excluded lipid levels which were so high that they could represent a primary lipid disorder per the NHLBI summary report.
Comment 11: Methods; in statistical way, how was the referent group in the logistic regression model considered? The borderline could also include the affected group from the preventative insight.
Response 11: TC, LDL-C, and levels which were above the abnormal threshold and HDL levels which were below the abnormal threshold are consistent with the diagnosis of dyslipidemia. Lipid levels which are within the borderline category do not meet the diagnostic criteria for dyslipidemia. We wanted to directly study the effect of dyslipidemia on our primary outcome and therefore used the abnormal threshold as the referent group.
Comment 12: Results; information of used drugs, especially steroids, should be added.
Response 12: The patient medications are not collected for this particular multicenter database and are therefore unknown.
Comment 13: The drugs should be added for multivariate analyses.
Response 13: The patient medications are not collected for this particular multicenter database and are therefore unknown.
Comment 14: Each Table; the abbreviations could be disclosure in footnote.
Response 14: Abbreviations added to footnote of each table: TC = total cholesterol, LDL-C = low density lipoprotein cholesterol, TG = triglyceride, HDL-C = high density lipoprotein cholesterol.
Comment 15: Discussion; the discussion of mechanistic explanation of results by glomerular etiology could be more detailed. The impact of etiology on the future cardiovascular diseases could be affected and the strategy might be altered by etiology.
Response 15: Added to discussion: ‘One possible explanation of why patients with glomerular disease have worse dyslipidemia is from impaired uptake of LDL-C by the LDL receptors in the mesangium of the glomerulus, leading to increased circulating LDL-C. When studied in a rat model, it was found that glycation of LDL-C impaired its ability to bind to LDL receptors in rats with diabetic kidney disease, which one type of glomerular disease. This mechanism may extend to other types of glomerular kidney disease and explain this observed difference between pediatric patients receiving dialysis with different etiologies to their kidney disease [18].
Comment 16: Discussion; as stated in the study limitation, the sampling methods for lipids were an issue. In fact, how large was the results of the study affected by this limitation? Triglycerides and LDL-cholesterol (if the calculation of LDL-cholesterol included the non-fasting triglyceride levels) could be concretely described.
Response 16: Given this is retrospective data, it is unknown whether or not the lipid levels were obtained from fasting or non-fasting samples.
Comment 17: Discussion; as stated in the study limitation, the explanation on the correlation between albumin and proteinuria should be described with any references (please cite evidence).
Response 17: Reference added: [23] Soeters, P. B.; Wolfe, R. R.; Shenkin, A. Hypoalbuminemia: Pathogenesis and Clinical Significance. Journal of Parenteral and Enteral Nutrition 2018, 43 (2), 181–193. DOI:10.1002/jpen.1451.
Comment 18: Line 14 (Abstract); patients’ demographics/attributes (i.e., mean age, gender prevalence) could be added.
Response 18: Added to abstract: ‘The cohort had a mean age was 15.2 years and were 54.5% female.’
Comment 19: Line 16; the word ‘being’ might be added after the expression ‘classified as’.
Response 19: The word ‘being’ was added after the expression 'classified as’.
Comment 20: Line 19; a full spell-out could be required for BMI as abbreviation.
Response 20: ‘body mass index’ was added to line 16, which is the first instance where BMI is used. Abbreviation also added to abbreviation list.
Comment 21: The expression ‘higher’ triglyceride in Line 22, ‘abnormal’ triglyceride in Line 24, and T ‘elevated’ triglycerides in Line 26; these appeared to be in a similar context, and the word ‘level’ may be added (e.g., higher level of triglyceride, or higher triglyceride level).
Response 21: Replaced ‘abnormal’ with ‘elevated’ or ‘decreased’ across manuscript. When using ‘abnormal’ additional clarification was added as ‘elevated TC, elevated LDL-C, elevated TG, and decreased HDL-C’. The term ‘level(s)’ was also added for clarity.
Comment 22: Line 38; a wide space was seen before ‘This’.
Response 22: wide space removed.
Comment 23: Line 106; a full spell-out could be required for BMI as abbreviation.
Response 23: ‘body mass index’ was added to line 16, which is the first instance where BMI is used. Abbreviation also added to abbreviation list.
Comment 24: Line 111; the word of ‘Lipids’ was written in an uppercase/large letter style. There were the same expressions in other parts (for instance, Line 232 etc.). It seemed to be tricky.
Response 24: Lipid(s) now in lowercase unless used as the first word in a sentence or as the first word in a table header.
Comment 25: Line 245; a wide space was seen before ‘Although’.
Response 25: wide space removed.
Comment 26: Line 265: ‘,’ might be added after ‘initiation’.
Response 26: ‘,’ added after ‘initiation’.
Reviewer 2 Report
Comments and Suggestions for Authors
This study focuses on the special group of pediatric dialysis patients and deeply explores the impact of glomerular diseases on their dyslipidemia, which is of certain significance. However, there are many places in the text that need to be revised and improved. For example, the introduction part should be appropriately compressed. The age range of the selected samples in the text is 0 - 21 years old. Different age groups may have different impacts on the results. This article selects those who received maintenance hemodialysis or peritoneal dialysis treatment between 2001 and 2016. The treatment 8 years ago is different from the current treatment. Charts can be appropriately added to the results to more intuitively show the statistical results. The format of references should be unified and mainly focus on the past five years.
Author Response
Comment 1: The introduction part should be appropriately compressed.
Response 1: Multiple portions of introduction have been removed as shown with strikethrough on the updated document.
Comment 2: The age range of the selected samples in the text is 0 - 21 years old. Different age groups may have different impacts on the results.
Response 2: We appreciate this comment and hope that our use of age-defined diagnosis of dyslipidemia in the logistic regressions helps to account for this. Our work shows a consistency between the findings of the linear regression which adjusts for age and the logistic regression which specifically assesses for age-defined dyslipidemia.
Comment 3: This article selects those who received maintenance hemodialysis or peritoneal dialysis treatment between 2001 and 2016. The treatment 8 years ago is different from the current treatment. Charts can be appropriately added to the results to more intuitively show the statistical results.
Response 3: We appreciate this comment and acknowledge the age of the cohort. Still, it is the most contemporary study of lipids in pediatric dialysis patients and can serve as a comparison study for changes overtime.
Comment 4: The format of references should be unified and mainly focus on the past five years.
Response 4: All reference updated for American Chemical Society (ACS) style and additional references added.
Round 2
Reviewer 1 Report
Comments and Suggestions for Authors
The revision might be still insufficient for being a high-quality paper. The points to be addressed (e.g., lack of information including steroid use and lipid assay/measurement way) should be described, including the effect on the results, in the limitations. In Line 123, here, triglyceride could be abbreviated.
Author Response
Comment 1: lack of information including steroid use and lipid assay/measurement way) should be described, including the effect on the results, in the limitations
Response 1: Added to discussion: ‘Although many patients discontinue immunosuppressants once they have reached ESKD, the impact of treatments on lipid levels may linger and may serve as a potential mechanism for the differences in lipids by disease type. Further studies which account for prior treatments are needed to evaluate medication use as an additional mediator.’
Comment 2: In Line 123, here, triglyceride could be abbreviated
Response 2: ‘Triglyceride’ abbreviated to ‘TG’
Reviewer 2 Report
Comments and Suggestions for Authors
This study focuses on the special group of pediatric dialysis patients and deeply explores the impact of glomerular disease on their dyslipidemia, which is of certain significance. However, there are several parts in the text that need to be revised and improved. For example, the introduction should be appropriately compressed. The age range of the selected samples in the text is 0 - 21 years old, and different age groups may have different effects on the results, so it is advisable to divide it into intervals. This article selected those who received maintenance hemodialysis or peritoneal dialysis treatment between 2001 and 2016. There have been significant differences in treatment between 8 years ago and now. Charts can be appropriately added to the results to present the statistical results more intuitively. The format of the references should be unified, and those published in the past five years should be mainly used.
Author Response
Comment 1: the introduction should be appropriately compressed.
Response 1: The length of the introduction has been shortened by more than 50%.
Comment 2: The age range of the selected samples in the text is 0 - 21 years old, and different age groups may have different effects on the results, so it is advisable to divide it into intervals.
Response 2: Median lipid levels by sex and age are shown in Supplemental Table S2 and S3. Analysis was performed for age less than 10 years old and 10 years or older. We hope that our use of age-defined diagnosis of dyslipidemia in the logistic regressions helps to account for this. Our work shows a consistency between the findings of the linear regression which adjusts for age and the logistic regression which specifically assesses for age-defined dyslipidemia.
Comment 3: This article selected those who received maintenance hemodialysis or peritoneal dialysis treatment between 2001 and 2016. There have been significant differences in treatment between 8 years ago and now.
Response 3: We appreciate this comment and acknowledge the age of the cohort. Still, it is the most contemporary study of lipids in pediatric dialysis patients and can serve as a comparison study for changes overtime.
Comment 4: Charts can be appropriately added to the results to present the statistical results more intuitively.
Response 4: Figure 1 added to manuscript to provide a more intuitive display of results.
Comment 5: The format of the references should be unified, and those published in the past five years should be mainly used.
Response 5: Reference list has been updated to include more recently published sources. 22% of references from the prior version of the manuscript were from past five years and 61% of references from this updated version of the manuscript were from past five years.